# Cinobufagin Exerts Anticancer Activity in Oral Squamous Cell Carcinoma Cells through Downregulation of ANO1

**DOI:** 10.3390/ijms222112037

**Published:** 2021-11-07

**Authors:** Sungwoo Jo, Eunhee Yang, Yechan Lee, Dongkyu Jeon, Wan Namkung

**Affiliations:** College of Pharmacy, Yonsei Institute of Pharmaceutical Sciences, Yonsei University, 85 Songdogwahak-ro, Yeonsu-gu, Incheon 21983, Korea; dsdyu2005@naver.com (S.J.); popcorncars@naver.com (E.Y.); llyycc94@naver.com (Y.L.); armisael1990@gmail.com (D.J.)

**Keywords:** cinobufagin, anoctamin 1, oral squamous cell carcinoma, CAL-27, STAT3

## Abstract

Anoctamin1 (ANO1), a calcium-activated chloride channel, is frequently overexpressed in several cancers, including oral squamous cell carcinoma (OSCC). OSCC is a highly aggressive cancer and the most common oral malignancy. ANO1 has been proposed as a potential candidate for targeted anticancer therapy. In this study, we performed a cell-based screening to identify novel regulators leading to the downregulation of ANO1, and discovered cinobufagin, which downregulated ANO1 expression in oral squamous cell carcinoma CAL-27 cells. ANO1 protein levels were significantly reduced by cinobufagin in a dose-dependent manner with an IC_50_ value of ~26 nM. Unlike previous ANO1 inhibitors, short-term (≤10 min) exposure to cinobufagin did not alter ANO1 chloride channel activity and ANO1-dependent intestinal smooth muscle contraction, whereas long-term (24 h) exposure to cinobufagin significantly reduced phosphorylation of STAT3 and mRNA expression of ANO1 in CAL-27 cells. Notably, cinobufagin inhibited cell proliferation of CAL-27 cells expressing high levels of ANO1 more potently than that of ANO1 knockout CAL-27 cells. In addition, cinobufagin significantly reduced cell migration and induced caspase-3 activation and PARP cleavage in CAL-27 cells. These results suggest that downregulation of ANO1 by cinobufagin is a potential mechanism for the anticancer effect of cinobufagin in OSCC.

## 1. Introduction

Oral squamous cell carcinoma (OSCC) is one of the most common head and neck cancers, accounting for 90% of all oral cancer types found in the mouth, tongue and lips [1]. According to GLOBOCAN 2020 reports, OSCC ranks sixth in overall cancer mortality and has a higher age-standardized mortality rate than pancreatic cancer [2]. Several topical and systemic treatments have been proposed to manage OSCC, but clinical results are poor [3]. To date, several anticancer targets for OSCC have been proposed. For example, HDAC8 is overexpressed in OSCC tissues and mainly localized in the cytoplasm. Downregulation of HDAC8 by siRNA in OSCC cell lines, YD-8, YD-10B and SNU-1076, inhibited cell proliferation and led to apoptotic cell death induction through caspase activation and pro-survival autophagy [4]. In OSCC cell lines, HN22 and HSC4, a reduction in specificity protein 1 (Sp1) expression by 2,4-bis (p-hydroxyphenyl)-2-butenal (HPB242) showed anti-proliferative effects [5]. GAP SH3 Binding Protein 1 (G3BP1) is highly expressed in metastatic OSCC, and downregulation of G3BP1 by siRNA significantly increased programmed cell death in the late stage p53-mutant OSCC [6]. GRB2 and IGF1 have identified as novel anticancer targets for OSCC using a series of bioinformatic approaches, including microarray analysis, network analysis and virtual screening [7]. However, due to the lack of effective drug targets for the treatment of OSCC, new potential drug targets for OSCC are needed.

Anoctamin1 (ANO1), also called transmembrane member 16A (TMEM16A), is known as a calcium-activated chloride channel [8,9,10]. It is expressed highly throughout the gastrointestinal tract and regulates various physiological activities by expression in smooth muscle, epithelial cells, small sensory neurons and olfactory sustentacular cells [11,12,13,14]. The ANO1 gene resides within the chromosome 11q13 region, which is frequently amplified in human cancers, and is associated with a poor prognosis [15,16,17,18]. ANO1 is involved in epithelial tumor formation and is highly amplified and expressed in OSCC [19], head and neck squamous cell carcinoma [20], prostate cancer [21], breast cancer [22] and esophageal cancer [23]. In particular, there is a very strong correlation between the amplification and expression of ANO1, and the overall survival rate of head and neck squamous cell carcinoma patients with high ANO1 protein expression is poor [24]. Notably, downregulation of ANO1 potently reduced the cell proliferation, metastasis and invasion in several cancer cells [25]. Previous studies revealed that inhibition of ANO1 can induce apoptosis through multiple signaling pathways including the NF-κB [26], TGF-β [27], CaMKII/MAPK [22] and EGFR/MAPK signaling pathway in cancer cells expressing ANO1 [28].

In previous studies, we identified several potent inhibitors of ANO1, such as Ani9, Ani9-5f, Ani-D2, luteolin and idebenone, which potently inhibited the chloride channel activity of ANO1, and Ani9-5f, Ani-D2, luteolin and idebenone significantly reduced cell proliferation and migration and induced apoptosis in several cancer cells highly expressing ANO1 [29,30,31,32,33]. However, all of these inhibitors and other previous ANO1 inhibitors strongly block the chloride channel activity of ANO1, which plays a pivotal role in normal physiological function. For example, fast inhibition of ANO1 chloride channel activity leads to a rapid decrease in blood pressure through the relaxation of vascular smooth muscle and a rapid decrease in gastrointestinal motility via the inhibition of pacemaker activity in interstitial cells of Cajal (ICC) [34,35,36,37]. These results suggest that the slow downregulation of ANO1 may be more beneficial in the treatment of cancer patients because it may exhibit anticancer activity without significant changes in blood pressure and gastrointestinal motility by providing sufficient time to compensate for the decreased ANO1 channel activity.

In this study, we identified a novel natural product, cinobufagin, which induces downregulation of ANO1 without alteration of ANO1 channel activity, and investigated the anticancer effects of cinobufagin on the human OSCC cell line CAL-27.

## 2. Results

### 2.1. Identification of Novel Compounds That Downregulate ANO1

Natural products have been a productive source for producing lead compounds and therapeutic agents, and some natural products potently and directly block ANO activity [32,33,38,39]. Therefore, we performed a cell-based screening using a natural product library containing 730 natural products to identify novel compounds inducing downregulation of ANO1. CAL-27 cells endogenously and highly expressing ANO1 were stably transfected with iodide-sensitive YFP-F46L/H148Q/I152L. The CAL-27 cells were treated with 20 μM test compounds for 24 h and washed with PBS, then the effects of the natural products on ANO1 channel activity were measured using YFP fluorescence quenching assay (Figure 1A). As shown in Figure 1B, some active compounds significantly inhibited the YFP fluorescence decrease via I^-^ uptake through the activated ANO1 chloride channels by 100 μM ATP. We identified 22 hit compounds that reduced ANO1 channel activity >70% at 20 μM. Of the 22 hits, only cinobufagin, fangchinoline and lanatoside C reduced the expression of ANO1 without inhibiting ANO1 ion channel activity. As shown in Figure 1C, cinobufagin, fangchinoline and lanatoside C did not alter the ANO1 channel activity, which was completely blocked by Ani9-5f, a potent and selective ANO1 inhibitor. The chemical structures of hit compounds are shown in Figure 1D. The protein expression levels of ANO1 significantly reduced by Ani9-5f, cinobufagin, fangchinoline and lanatoside C in CAL-27 cells, and cinobufagin most strongly reduced ANO1 protein level. Thus, we performed further studies on ANO1 downregulation by cinobufagin.

### 2.2. Effect of Cinobufagin on Protein Expression Levels of ANO1 and CFTR

To investigate the effect of different concentrations of cinobufagin on ANO1 protein expression levels, Western blot analysis was performed in CAL-27. As shown in Figure 2A,B, cinobufagin reduced ANO1 protein expression levels in a dose-dependent manner. ANO1 protein expression levels were reduced by 12.7%, 45.5%, 66.8% and 67.2% by 10, 30, 100 and 300 nM of cinobufagin, respectively.

In order to determine whether ANO1 protein expression level is selectively reduced by cinobufagin, we investigated the effect of cinobufagin on the protein expression level of another chloride channel, cystic fibrosis transmembrane conductance regulator (CFTR). FTR cells expressing CFTR were treated with different concentrations of cinobufagin for 24 h, and then Western blot analysis was performed to evaluate the effect of cinobufagin on the protein expression level of CFTR. Notably, high concentrations of cinobufagin did not alter the protein expression levels of CFTR (Figure 2C,D). These results suggest that cinobufagin may selectively reduce the protein expression level of ANO1.

### 2.3. Effect of Cinobufagin on ANO1 Acitivy and Intestinal Smooth Muscle Contration

Previous potent and selective ANO1 inhibitors, such as Ani9-5f, strongly block the chloride channel activity of ANO1 [29,30]. Rapid inhibition of ANO1 chloride channel activity leads to decreased blood pressure and gastrointestinal motility through induction of smooth muscle relaxation and inhibition of pacemaker activity of ICC [34,35,36,37]. Therefore, these physiological actions of previous ANO1 inhibitors are likely to act as an obstacle in the development of anticancer therapies targeting ANO1. An electrophysiological study was performed to more precisely investigate the effect of cinobufagin on ANO1 channel activity. Apical membrane currents of ANO1 were measured by Ussing chamber technique in FRT cells expressing wild-type human ANO1. As shown in Figure 3A, short-term (10 min) pretreatment with cinobufagin had no inhibitory effect on ANO1 chloride channel activity, even at high concentrations, but Ani9-5f completely blocked ANO1 chloride channel activity.

To determine whether cinobufagin affects smooth muscle contraction, the effect of cinobufagin on intestinal smooth muscle contraction was measured using mouse ileum. As shown in Figure 3B, cinobufagin did not significantly reduce the spontaneous contraction of mouse ileum, but a potent ANO1 inhibitor, Ani9-5f, strongly and almost completely reduced intestinal smooth muscle contraction, as expected.

### 2.4. Cinobufagin Reduced the Phosphorylation of STAT3 and ANO1 Gene Transcription

Cinobufagin potently reduced ANO1 protein expression in a dose-dependent manner (Figure 2A). To confirm whether this phenomenon of cinobufagin was achieved through the regulation of ANO1 mRNA expression, real-time PCR was performed. Interestingly, cinobufagin significantly reduced the mRNA expression level of ANO1 in a dose-dependent manner in CAL-27 cells. ANO1 mRNA expression levels were reduced by 8.6, 37.9, 47.8, 69.4 and 82.1% by 10, 30, 100, 300 and 1000 nM of cinobufagin, respectively.

Recent studies suggest that ANO1 gene transcription regulated by signal transducer and activator of transcription (STAT) transcription factors, including STAT3 [40]. In addition, STAT3 is recognized as an oncogene, and its activity is increased by ~50% in various cancers, including OSCC [41,42,43,44]. To investigate whether cinobufagin affects ANO1 expression via regulation of STAT3 activity, Western blot analysis was performed in CAL-27 cells. Interestingly, cinobufagin significantly blocked STAT3 phosphorylation in a dose-dependent manner. The phosphorylation of STAT3 reduced by 3.9, 25.5, 49 and 78.5% by 10, 30, 100 and 300 nM of cinobufagin (Figure 4B). Western blot analysis was performed in CAL-27 cells treated with niclosamide, an inhibitor of STAT3, to observe whether inhibition of STAT3 reduced ANO1 expression. As expected, niclosamide significantly reduced STAT3 phosphorylation (Figure 4D,E) and mRNA expression levels of ANO1 (Figure 4F) in CAL-27 cells. These results suggest that cinobufagin may reduce mRNA expression levels of ANO1 via inhibition of STAT3 pathway.

### 2.5. Cinobufagin Reduced Cell Proliferation and Migration in CAL-27 Cells

To investigate the effect of the cinobufagin-induced downregulation of ANO1 on cell viability, the ANO1 knockout CAL-27 cell line was established by CRISPR-Cas9. Of interest, cinobufagin more potently reduced cell viability of ANO1 expressing CAL-27 cells compared with ANO1 knockout CAL-27 cells (Figure 5A). In CAL-27 cells highly expressing ANO1, cell viability was significantly reduced by 24.2, 44.0, 56.7 and 69.0% by 30, 100, 300 and 1000 nM of cinobufagin, respectively. However, in ANO1 knockout CAL-27 cells, cinobufagin much less potently reduced cell viability by 19.4, 30.3, 44.4 and 57.7% by 30, 100, 300 and 1000 nM of cinobufagin, respectively (Figure 5A).

To determine whether cinobufagin reduces cell migration, a scratch wound assay was performed on CAL-27 cells expressing high levels of ANO1. As shown in Figure 5B,C, cell migration was strongly reduced by cinobufagin in a dose-dependent manner.

### 2.6. Cinobufagin-Induced Activation of Caspase-3 and Cleavage of PARP

A number of previous studies have shown that inhibition or downregulation of ANO1 induces apoptosis in various cancer cell lines [32,33,45,46,47]. Here, we explore whether cinobufagin induces apoptosis in CAL-27 cells highly expressing ANO1. As shown in Figure 6A,B, cinobufagin treatment significantly increased caspase-3-positive cells compared with control. The cinobufagin-induced increase in caspase-3-positive cells was almost completely inhibited by Ac-DEVD-CHO, a caspase-3 inhibitor. In addition, cinobufagin treatment induced a significant increase in cleaved PARP-1, considered a marker of apoptosis, in a dose-dependent manner in CAL-27 cells (Figure 6C,D).

## 3. Discussion

Recent studies have suggested ANO1 as a potential therapeutic target for several cancers, including OSCC [19,20,21,22,23]. To date, several ANO1 inhibitors have been developed through the efforts of many researchers. However, it has not yet led to the development of a therapeutic agent. Natural products have been an efficient and productive source of lead compound generation in drug discovery [38]. In the present study, we have identified a novel and potent natural product, cinobufagin, as a potential therapeutic agent for OSCC which may have anticancer effects on other cancers that target ANO1. Cinobufagin is one of the main components of Chansu, a traditional Chinese medicine obtained from the secreted substance of Bufo bufo gargarizans (a toad) [48]. Interestingly, cinobufagin reduced the ANO1 protein expression level by ~45% at 30 nM and significantly inhibited the cell viability of the OSCC cell line CAL-27 at >30 nM (Figure 2B and Figure 5A). In general, natural products act on the target proteins in the micromolar range, and the effect is rarely seen at less than 1 μM [49].

Previous studies have shown that cinobufagin reduced the expression of LEF1 and Wnt/β-catenin target genes such as Axin-2, cyclin D1, and c-Myc in melanoma cell lines [50], and cinobufagin treatment reduced the expression of p-AKTT308 and p-AKTS473 and inhibited the AKT/mTOR signaling pathway in human non-small cell lung cancer (NSCLC) cells [51]. In addition, cinobufagin effectively induced apoptosis in A549 cells by triggering caspase activation through both intrinsic and extrinsic pathways [52]. In colorectal cancer, cinobufagin inhibited proliferation, migration, invasion and promoted apoptosis of HCT116, RKO and SW480 cells. This is the result of cinobufagin inhibiting the epithelial-mesenchymal transition in colorectal cancer by inhibiting the STAT3 pathway [53]. In breast cancer, anti-proliferative and pro-apoptotic effects of punicalagin were confirmed in MCF-7 cells [54]. These results show that several mechanisms have been proposed for the anticancer effect of cinobufagin. However, in Figure 5A, we showed that high concentrations of cinobufagin also reduced the cell viability of ANO1 knockout CAL-27 cells, and ANO1-dependent reduction in cell viability was further significantly reduced in CAL-27 cells highly expressing ANO1. These results suggest that the pathway that reduces ANO1 expression by cinobufagin is involved, at least in part, in the mechanism of its anticancer activity. In addition, ANO1 is highly expressed in HCT116, SW480 and MCF-7 cancer cells [47,55], where cinobufagin showed anticancer activity in previous studies. Anticancer effects such as decreased cell viability due to downregulation of ANO1 are shown in these cancer cells [47,55]. Therefore, it is possible that the downregulation of ANO1 by cinobufagin acted as an important anticancer mechanism in these cancer cells.

Since ANO1 is a calcium-activated chloride channel that plays a pivotal role in the regulation of important physiological functions, other physiological functions of ANO1 should also be considered in the development of anticancer drugs targeting ANO1. Previous small molecule inhibitors of ANO1 rapidly and potently block ANO1 chloride channel activity, although they show anticancer activity, including a reduction in cell proliferation, metastasis and invasion [21,31,39,45,46,47,50]. However, when these ANO1 inhibitors are used in patients, there is a risk of side effects such as a drop in the patient’s blood pressure or inhibition of intestinal motility due to the rapid inhibition of ANO1 chloride channel activity [34,35,36,37]. One way to reduce these side effects is to slowly decrease the chloride channel activity of ANO1, providing sufficient time to compensate for the decreased ANO1 chloride channel activity. Cinobufagin, unlike other ANO1 inhibitors, does not directly alter the chloride channel activity of ANO1 (Figure 3A) and gradually reduces the expression level of ANO1 (Figure 2A and Figure 4A). Thus, it has the potential to be developed as an anticancer agent that can reduce side effects and exhibit anticancer effects.

Our study showed an efficient method for screening modulators that downregulate ANO1 without affecting the chloride channel activity of ANO1 and suggested that the downregulation of ANO1 by small molecule modulator is useful for the development of ANO1 targeting anticancer agents.

## 4. Materials and Methods

### 4.1. Material and Solutions

The compound collections used for HTS included 730 natural products from Target Molecule Corp. (Wellesley Hills, MA, USA). Cinobufagin was purchased from MedChem Express (Monmouth Junction, NJ, USA). Lanatoside C and fangchinoline were purchased from Sigma-Aldrich (St. Louis, MO, USA). HEPES buffer solution contains 140 mM NaCl, 5 mM KCl, 1 mM MgCl_2_, 1 mM CaCl_2_, 10 mM D-glucose and 10 mM HEPES (pH 7.4) The HCO_3_^−^-buffered solution contains 140 mM NaCl, 5 mM KCl, 1 mM MgCl_2_,1 mM CaCl_2_, 10 mM D-glucose, 2.5 mM HEPES and 25 mM NaHCO_3_ (pH 7.4).

### 4.2. Cell Culture

CAL-27 were cultured in Dulbecco’s modified Eagle medium (DMEM). DMEM was supplemented with 10% FBS, 100 U/mL penicillin, and 100 μg/mL streptomycin. FRT cell lines stably expressing CFTR and ANO1 were provided by Alan Verkman (University of California, Los Angeles, CA, USA) and cultured in F-12 Modified Coon’s medium with 10% fetal bovine serum (FBS), 100 U/mL penicillin, 100 μg/mL streptomycin and 2 mM L-glutamine. All cells were grown at 37 °C, 5% CO_2_ and 95% humidity.

### 4.3. Cell Based YFP Fluorescence Quenching Assay

CAL-27 cells stably expressing halide sensor YFP-F46L/H148Q/I152L were seeded in 96-well microplates at a confluence of ~80% per well and incubated for 24 h. Test compounds were treated in each well at a concentration of 20 μM, and then incubated for 24 h. Then, each well of the 96-well plate was washed twice with PBS and incubated with 100 μL of HEPES buffer solution for 10 min at 37 °C. After incubation, YFP fluorescence in each well was monitored for 10 s and recorded every 0.4 s with the FLUOstar Omega microplate reader (BMG Labtech, Ortenberg, Germany), and baseline fluorescence was measured for the first 0.8 s. Then, at 1 s, iodide HEPES buffer solution (NaI replacing NaCl) containing 100 μM ATP was applied using a syringe pump to measure iodide influx-related ANO1 activity. The effect of ANO1-mediated iodide influx by test compounds was measured by the initial slope value of YFP fluorescence.

### 4.4. Apical Membrane Circuit Measurement

FRT cells stably expressing ANO1 were seeded at a confluence of 3 × 10^5^ cells/cm^2^ on snapwell inserts (1.12 cm^2^ surface area) and cultured until confluent. Snapwell inserts were mounted in Ussing chambers (Physiologic Instruments, San Diego, CA, USA). Basolateral was bathed with HCO_3_^−^-buffered solution and apical was bathed with half chloride HCO_3_^−^-buffered solution (70 mM Na gluconate replacing 70 mM NaCl). Cells were stabilized for 40 min, bathing in a buffered solution aerated with 95% O_2_, 5% CO_2_ at 37 °C. Cinobufagin was treated with both apical and basolateral bath solutions 10 min before ANO1 activation. Then, 100 μM of ATP was loaded to the apical bathing solution to activate ANO1. Apical membrane and short-circuit currents were measured with a Power Lab 4/35 (AD Instruments, Castle Hill, Australia) and EVC4000Multi-Channel V/I Clamp (World Precision Instruments, Sarasota, FL, USA). Data were evaluated using Lab chart Pro 7 (AD Instruments, Castle Hill, Australia). The sampling rate was 4 Hz.

### 4.5. Immunoblot Analysis

Protein sample preparation protocol was described previously [33]. Protein samples were separated using 4–12% Tris Glycine Precast Gel (KOMA BIOTECH, Seoul, Korea). Then, separated proteins were transferred to polyvinylidene fluoride (PVDF) membranes. Blocking was carried out using Tris-buffered saline with 0.1% Tween 20 (TBST) containing 5% nonfat skim milk or 5% BSA at room temperature for 1 h. Then, the membranes were incubated with primary antibodies, including anti-TMEM16A antibody [SP31] (Abcam, Cambridge, UK), CFTR [M3A7] (Sigma-Aldrich, St. Louis, MO, USA), phospho-STAT3 [EP2147Y] (Abcam, Cambridge, UK), STAT3 [#9139] (cell signaling), anti-cleaved PARP (BD Biosciences) and anti-β-actin (Santa Cruz Biotechnology, Dallas, TX, USA) overnight at 4 °C. Subsequently, the membranes were washed out with TBST 3 times at 5 min intervals and incubated with HRP-conjugated anti-secondary IgG antibodies (Santa Cruz Biotechnology, Dallas, TX, USA) for 1 h at room temperature. Finally, visualization was carried out with the SuperSignal™ Western Blot Substrate (Thermo Fisher Scientific, Waltham, MA, USA)

### 4.6. Real-Time RT-PCR Analysis

TRIzol solution (Invitrogen, Carlsbad, CA, USA) was used to extract total mRNA. Total mRNA was reverse transcribed using an oligo (dT) primer, random hexamer primers and SuperScript III Reverse Transcriptase (Invitrogen, Carlsbad, CA, USA). Thunderbird SYBR qPCR mix (Toyobo, Osaka, Japan) and StepOnePlus Real-Time PCR System (Applied Biosystems, Waltham, MA, USA) were used for quantitative RT PCR. The thermal cycling conditions consisted of 95 °C for 5 min, 40 cycles of 95 °C for 10 s, 60 °C for 20 s and 72 °C for 10 s. The ANO1 sense primer sequence is 5′-GGAGAAGCAGCATCTATTTG-3′ and the ANO1 antisense primer sequence is 5′-GATCTCATAGACAATCGTGC-3′. The size of the ANO1 PCR product is 82 base pairs.

### 4.7. Cell Viability Assay

Cell Titer 96^®^ AQueous One Solution Assay kit (MTS) (Promega, Madison, WI, USA) was used for performing cell viability assay. CAL-27 cells were cultured in 96-well plates with growth medium supplemented with 10% FBS for 24 h. When cell density reached ~ 40%, dimethyl sulfoxide solution (vehicle) and cinobufagin were treated in medium, exchanged freshly every 12 h. After 48 h treatment, the medium was washed out and MTS assay was conducted by the supplier’s instructions. The absorbance was measured by Infinite M200 microplate reader (Tecan, Männedorf, Switzerland) at a wavelength of 490 nm.

### 4.8. Wound Healing Assay

CAL-27 cells were cultured with approximately 100% confluence to form a monolayer in a 96-well plate. Wound area was formed by using 96-Well Wound Maker (Essen BioScience, Ann Arbor, MI, USA). Then, cells were washed out twice with PBS and incubated with 200 μL of growth medium containing 1% FBS with cinobufagin or vehicle (DMSO). Images of the wound area were taken using IncuCyte ZOOM (Essen BioScience, Ann Arbor, MI, USA), and the percentage of wound closure was analyzed using IncuCyte software.

### 4.9. Caspase-3 Activity Assay

CAL-27 cells were cultured in 96-well plates at a density of ~40%. Then, each well was treated with cinobufagin and a caspase-3 inhibitor, Ac-DEVD-CHO, for 24 h. Then, each well was washed out twice with 100 μL PBS and incubated for 30 min in 100 μL of PBS with 1 μM of caspase-3 substrate, NucView 488, at room temperature. After incubation, 1 μM Hoechst 33342 was added to stain the cells. FLUOstar Omega microplate reader (BMG Labtech, Ortenberg, Germany) was used to measure the fluorescence of Hoechst 33342, and NucView 488 and Lionheart FX Automated Microscope (BioTek, Winooski, VT, USA) were used to capture multicolor images.

### 4.10. Intestinal Smooth Muscle Contraction

C57BL/6 mice were killed by CO_2_ euthanasia at age 8–10 weeks. The animal study protocols were approved by the Institutional Animal Ethics Committee of Yonsei University. The harvested ileum was washed with cold HCO_3_^-^ buffer solution. Then, the ileal segments were fixed to the silk string, connected to the force transducer and stabilized until the tension reached ~1 mN in 60 min. The bathing solution was changed at 15 min intervals. Tension was determined with a fixed-range precision force transducer (TSD, 125 C; Biopac, Goleta, CA, USA) connected to a differential amplifier (DA 100B; Biopac). MP100 and Biopac digital acquisition system were used to recording data and Acknowledge 3.5.7 software (Biopac) was used to analyze data.

### 4.11. Statistical Analysis

All experiments were performed independently for a minimum of three times. Statistical analyses were performed using GraphPad Prism 5.0 (GraphPad Software Inc., San Diego, CA, USA). The results for multiple trials are presented as the mean ± standard deviation (S.D.). Student’s *t*-test or analysis of variance was used performing statistical analysis. Statistical significance was set at *p* values less than 0.05.

## Figures and Tables

**Figure 1 ijms-22-12037-f001:**
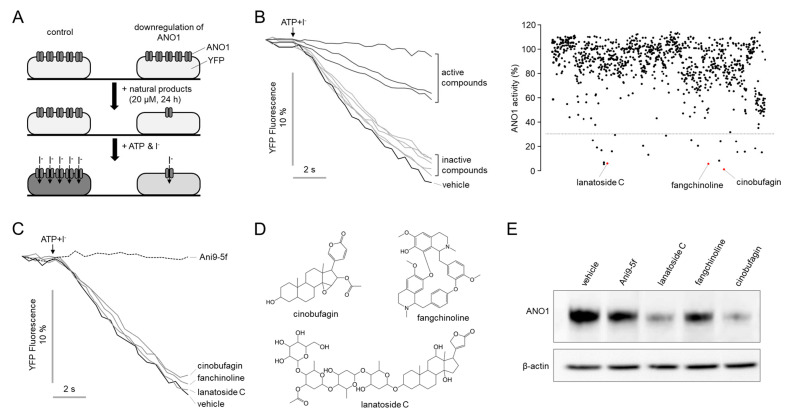
Identification of novel compounds that downregulate ANO in CAL-27 cells. (**A**) Principle of high-throughput screening assay. (**B**) Representative YFP fluorescence traces. YFP fluorescence monitored in CAL-27 cells treated with 20 μM test compounds for 24 h. (right) Dot plot of primary screening results for 730 natural products. (**C**) Representative YFP fluorescence traces showing the effect of short-term (10 min) exposure of 20 μM cinobufagin, fangchinoline and lanatoside C on ANO1 channel activity in CAL-27 cells. (**D**) Chemical structures of cinobufagin, fangchinoline and lanatoside C. (**E**) Effects of Ani9-5f, cinobufagin, fangchinoline and lanatoside C on ANO1 protein expression levels were assessed by Western blot analysis. CAL-27 cells were treated with 10 μM of the test compounds for 24 h.

**Figure 2 ijms-22-12037-f002:**
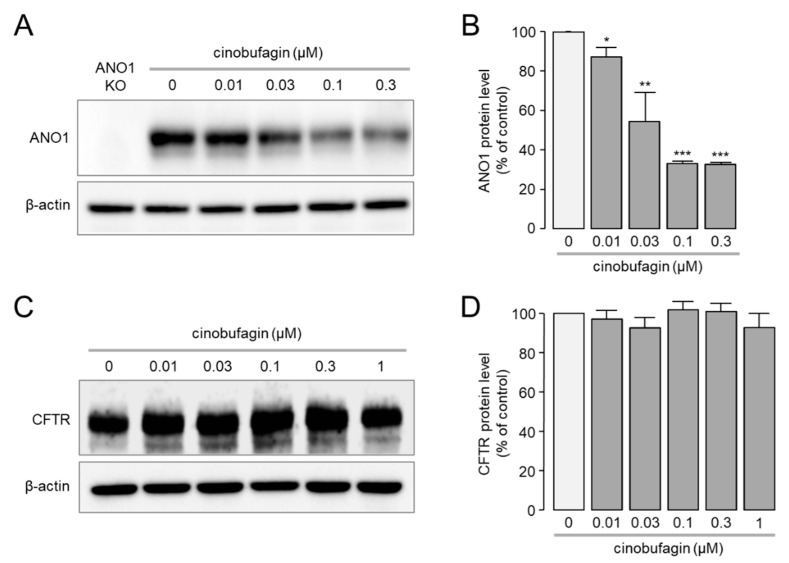
Effect of cinobufagin on protein expression level of ANO1 and CFTR. (**A**) Representative immunoblot analysis of ANO1 in cinobufagin-treated CAL-27 cells. Cinobufagin-treated cells at the indicated concentration for 24 h. CRISPR/Cas9 technique was used to generate CAL-27 ANO1 knockout (KO) cells. (**B**) ANO1 protein intensities were normalized to those of β-actin (mean ± S.D., *n* = 4). * *p* < 0.05, ** *p* < 0.01, *** *p* < 0.001. (**C**) Representative immunoblot blot analysis of CFTR in cinobufagin-treated FRT cells expressing human CFTR. Cinobufagin-treated cells at the indicated concentration for 24 h. (**D**) CFTR protein intensities were normalized to those of β-actin (mean ± S.D., *n* = 3).

**Figure 3 ijms-22-12037-f003:**
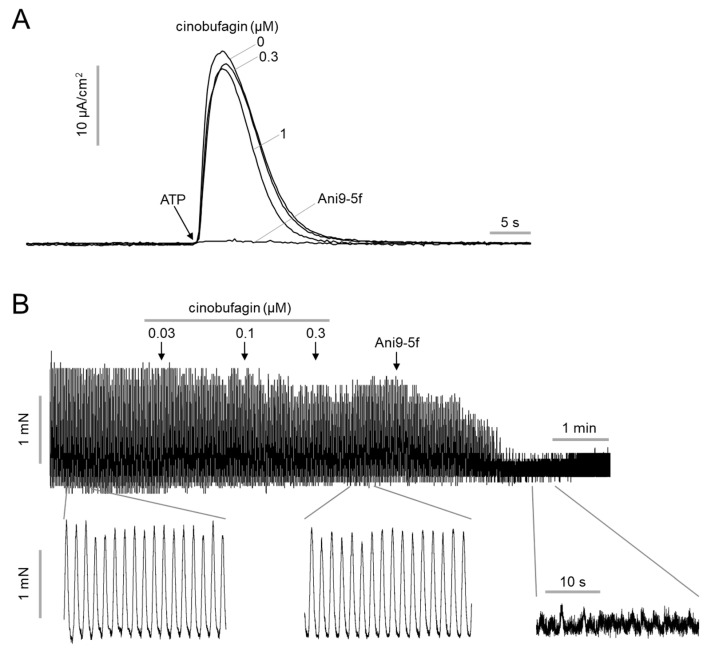
Effect of cinobufagin on ANO1 chloride channel activity and intestinal smooth muscle contraction. (**A**) Representative apical membrane currents from 3 independent experiments in FRT cells expressing ANO1. Indicated concentrations of cinobufagin and 10 μM Ani9-5f were pretreated for 10 min. ANO1 chloride channel was activated by 100 μM ATP. (**B**) Representative smooth muscle contraction traces from 4 independent experiments with mouse ileal segments. Indicated concentrations of cinobufagin and 10 μM Ani9-5f applied to mouse ileal segments.

**Figure 4 ijms-22-12037-f004:**
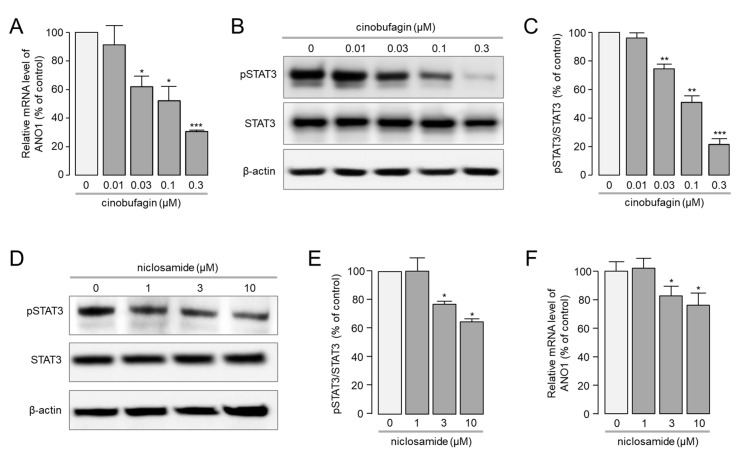
Effects of cinobufagin on mRNA expression level of ANO1 and phosphorylation of STAT3 in CAL-27 cells. (**A**) mRNA expression levels of ANO1 were measured using real-time PCR in CAL-27 cells. Indicated concentrations of cinobufagin were treated for 24 h (mean ± S.E., *n* = 3). (**B**) Representative immunoblot analysis of pSTAT3 in CAL-27 cells treated with cinobufagin for 24 h. (**C**) Phosphorylated STAT3 (pSTAT3) levels were normalized against total STAT3 levels (mean ± S.E., *n* = 3). (**D**) Representative immunoblot analysis of pSTAT3 in CAL-27 cells treated with niclosamide at indicated concentrations for 24 h. (**E**) pSTAT3 levels were normalized against total STAT3 levels (mean ± S.E., *n* = 3). (**F**) mRNA expression levels of ANO1 were measured in CAL-27 cells treated with niclosamide at indicated concentrations for 24 h (mean ± S.E., *n* = 3). * *p* < 0.05, ** *p* < 0.01, *** *p* < 0.001.

**Figure 5 ijms-22-12037-f005:**
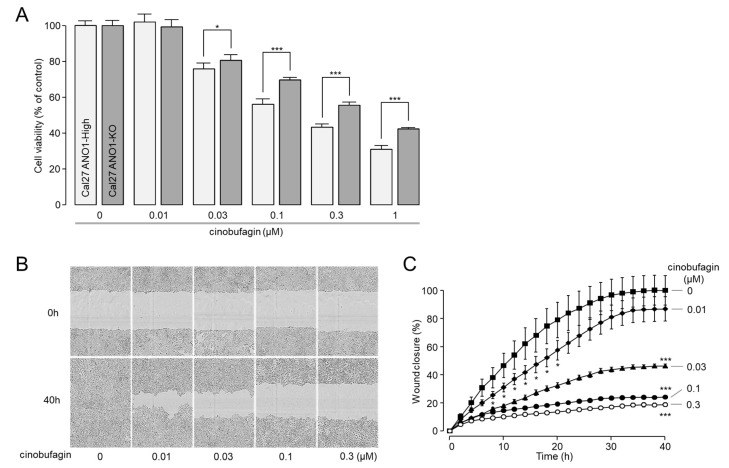
Effect of cinobufagin on cell viability and migration in CAL-27 cells. (**A**) Effect of cinobufagin on cell viability in CAL-27, and ANO1 knockout (KO) CAL-27 cells. These cells were treated with cinobufagin at the indicated concentrations for 48 h, and MTS assay was performed to estimate cell viability (mean ± S.E., *n* = 5). (**B**,**C**) Scratch wound healing assay was conducted in CAL-27 cells expressing high levels of ANO1. Cells were treated with cinobufagin at the indicated concentration, and time-lapse images were acquired every 2 h after wound generation (mean ± S.E., *n* = 3). Scale bars represent 300 μm. * *p* < 0.05, *** *p* < 0.001.

**Figure 6 ijms-22-12037-f006:**
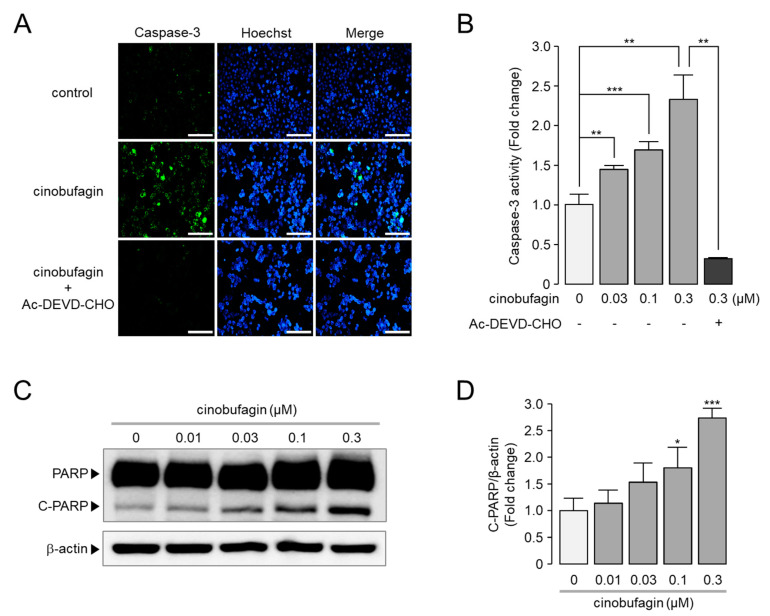
Effect of cinobufagin on caspase-3 activation and induction of PARP cleavage in CAL-27 cells. (**A**) Images were taken 24 h after application of 300 nM cinobufagin. Cells were incubated with caspase-3 substrate (green) and Hoechst 33342 (blue) 20 min prior to image acquisition. Scale bars represent 200 μm. (**B**) CAL-27 cells were treated with the indicated concentrations of cinobufagin in the presence or absence of 10 μM of Ac-DEVD-CHO for 24 h, then the cells were treated with 2 μM of caspase-3 substrate for 20 min to estimate caspase-3 activity (mean ± S.E., *n* = 3). (**C**) Representative immunoblot analysis of PARP and cleaved PARP(C-PARP). Cells were treated with the indicated concentrations of cinobufagin for 24 h. (**D**) Cleaved PARP protein intensities were normalized to those of β-actin (mean ± S.D., *n* = 3). * *p* < 0.05, ** *p* < 0.01, *** *p* < 0.001.

## Data Availability

Not applicable.

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
