# Peer review of "Cinobufagin Exerts Anticancer Activity in Oral Squamous Cell Carcinoma Cells through Downregulation of ANO1"

_ijms, 2021, doi:10.3390/ijms222112037_

Round 1

Reviewer 1 Report

An interesting original article showing the potential anticancerous effect of cinobufagin in Oral squamous Cell Carcinoma by exploiting its ability to downregulate Anoctamin 1.

Only minor queries :

Page 1 line 31 you should add: "Various topical and systemic treatments have been proposed to manage OSCC, with poor clinical results" and cite articles such as: doi: 10.3390/curroncol28040213. and doi: 10.3390/medicina57060563.

Page 11 line 363 4.11. Statistical Analysis

please specify what statistical program, version (also the program's maker and location) you used to calculate significance.

Thank You 

Author Response

We greatly appreciate the editor’s and reviewers’ efforts to carefully review our manuscript and the valuable comments and suggestions offered for the improvement of the manuscript (ijms-1431042). We have made each of the suggested revisions and carried out additional experiments as needed. The points of criticism raised by the reviewers were addressed by a point-by-point response. Changes in the manuscript text are highlighted in red color font.

Reviewer #1:

An interesting original article showing the potential anticancerous effect of cinobufagin in Oral squamous Cell Carcinoma by exploiting its ability to downregulate Anoctamin 1.

Only minor queries:

  1. Page 1 line 31 you should add: "Various topical and systemic treatments have been proposed to manage OSCC, with poor clinical results" and cite articles such as: doi: 10.3390/curroncol28040213. and doi: 10.3390/medicina57060563.

Response: Thank you for the valuable comments. It was described in the revised manuscript as suggested.

  1. Page 11 line 363 4.11. Statistical Analysis; please specify what statistical program, version (also the program's maker and location) you used to calculate significance.

Response: Thank you. Corrected.

Reviewer 2 Report

The manuscript ijms-1431042 entitled “Cinobufagin Exerts Anticancer Activity in Oral Squamous Cell Carcinoma Cells through Downregulation of ANO1” deals with detection and efficacy testing of natural compound – cinobufarin aimed at more effective approach in battling oral cancer cells.  

The manuscript is in the scope of the Journal, and some minor issues still remain before it could be accepted.

Please use data from GLOBOCAN to support your data about incidence, prevalence, and mortality of OSCC. Please define “high mortality rate”. E.g. Lip and oral cavity cancer has mortality ASR 10x lower than lung cancer (lines 28-30).

Why is important to emphasize GRB2 and IGF1 (line 41) as drug targets, since the whole paper is focused on ANO1?

Please mark subfigures in Figure 1 – A-F. 1B, 1B right, 1C, and 1D do not have required resolution. Note that axis in 1B right is broken. What would be x- and y- axes in 1B and C? What is plotted on x-axis in 1B right?

Lanatoside C, fangchinoline, and cinobufagin are marked in red, which compounds are visible next to Lanatoside C and why were they not considered?

Please add axes in Fig 3.

Can you describe briefly what is the source of cinobufagin in the Discussion?

Based on the statement in lines 254-256, is there a possibility that cancer cells would compensate these activities as well?

Would cinobufagin exert properties that enable its oral application to patients?

Please add full description of abbreviations, subscript O2 (line 298).

Author Response

We greatly appreciate the editor’s and reviewers’ efforts to carefully review our manuscript and the valuable comments and suggestions offered for the improvement of the manuscript (ijms-1431042). We have made each of the suggested revisions and carried out additional experiments as needed. The points of criticism raised by the reviewers were addressed by a point-by-point response. Changes in the manuscript text are highlighted in red color font.

  1. Please use data from GLOBOCAN to support your data about incidence, prevalence, and mortality of OSCC. Please define “high mortality rate”. E.g. Lip and oral cavity cancer has mortality ASR 10x lower than lung cancer (lines 28-30).

Response: Thank you for the valuable comments. It is described in the revised manuscript as suggested.

  1. Why is important to emphasize GRB2 and IGF1 (line 41) as drug targets, since the whole paper is focused on ANO1?

Response: Sorry for the confusion. We agree with the reviewer’s comments, and the sentence has been revised to emphasize ANO1.

  1. Please mark subfigures in Figure 1 – A-F. 1B, 1B right, 1C, and 1D do not have required resolution. Note that axis in 1B right is broken. What would be x- and y- axes in 1B and C? What is plotted on x-axis in 1B right?

Response: Thank you for the comments. We used the x and y axes to clearly show the changes of YFP fluorescence. We have used this format in more than 10 papers. The resolution of Figure 1 has been adjusted appropriately.

  1. Lanatoside C, fangchinoline, and cinobufagin are marked in red, which compounds are visible next to Lanatoside C and why were they not considered?

Response: Those compounds were excluded because they affected the ANO1 channel activity.

  1. Please add axes in Fig 3.

Response: Thank you. Corrected.

  1. Can you describe briefly what is the source of cinobufagin in the Discussion?

Response: Thank you for the valuable comments. As suggested, it was described in the discussion section of the revised manuscript. 

  1. Based on the statement in lines 254-256, is there a possibility that cancer cells would compensate these activities as well?

Response: Since most of the previous studies have shown short-term anticancer effects of ANO1 inhibition in animal models, we believe that cancer cells have the potential to compensate for ANO1 via novel pathways as reviewer's opinion. For example, it is possible to generate an ANO1 KO cancer cell line that grows similar to cancer cells overexpressing ANO1. This result supports the possibility.

  1. Would cinobufagin exert properties that enable its oral application to patients?

Response: In a previous study, pharmacokinetic analysis after oral administration of toad venom containing cinobufagin to SD rats showed the potential for development of cinobufagin as an oral drug. (J Pharm Biomed Anal. 2008 Feb 13;46(3):442-8. doi: 10.1016/j.jpba.2007.11.001)

  1. Please add full description of abbreviations, subscript O2 (line 298).

Response: Thank you. Corrected.

Reviewer 3 Report

This stud suggested that ANO1 is correlated with cinobufagin exerts induced anti-tumor effect. However, the major weakness of this paper is that they only confirm this effect on single cells and without in vivo study to support their finding. The data is relatively less to convince reader.

  1. The define of long-term and short-term exposure is unclear.
  2. In figure 1E, Ani9-5f may need to perform with Western of ANO1.
  3. In figure 4B, author suggested that cinobufagin suppress phosphorylation of STAT3. There is no evidence support that cinobufagin may suppress ANO1 mRNA level via inhibition of STAT3. Author overestimated their finding. To support this summary, author need to performed with siRNA of STAT3 to check the changed of ANO1.
  4. The statistic symbol need to be added onto figure 5c.

Author Response

We greatly appreciate the editor’s and reviewers’ efforts to carefully review our manuscript and the valuable comments and suggestions offered for the improvement of the manuscript (ijms-1431042). We have made each of the suggested revisions and carried out additional experiments as needed. The points of criticism raised by the reviewers were addressed by a point-by-point response. Changes in the manuscript text are highlighted in red color font.

Reviewer #3:

This study suggested that ANO1 is correlated with cinobufagin exerts induced anti-tumor effect. However, the major weakness of this paper is that they only confirm this effect on single cells and without in vivo study to support their finding. The data is relatively less to convince reader.

  1. The define of long-term and short-term exposure is unclear.

Response: Sorry for the confusion. In this study, short- and long-term exposures were used to emphasize the duration of drug treatment. The short-term (£10 minutes) and long-term (24 h) time ranges were described in the revised manuscript.

  1. In figure 1E, Ani9-5f may need to perform with Western of ANO1.

Response: Thank you for the valuable comments. As suggested, we performed Western blot analysis to show the effect of Ani9-5f (a positive control) on the protein expression level of ANO1. Interestingly, cinobufagin, fangchinoline and lanatoside C more potently reduced the ANO1 protein expression level compared to Ani9-5f (Figure 1E).

  1. In figure 4B, author suggested that cinobufagin suppress phosphorylation of STAT3. There is no evidence support that cinobufagin may suppress ANO1 mRNA level via inhibition of STAT3. Author overestimated their finding. To support this summary, author need to performed with siRNA of STAT3 to check the changed of ANO1.

Response: Thank you for the valuable comments. We performed Western blot analysis and real-time RT-PCR to investigate whether inhibition of STAT3 affected ANO1 expression in CAL-27 cells. As shown in Figure 4D-4F, inhibition of STAT3 by niclosamide, an inhibitor of STAT3, significantly reduced STAT3 phosphorylation and mRNA expression levels of ANO1 in CAL-27 cells.

  1. The statistic symbol need to be added onto figure 5c.

Response: Thank you. Corrected.

Round 2

Reviewer 3 Report

I have no further question for this work.